# VISIT: Visualizing and Interpreting the Semantic Information Flow of Transformers

**Shahar Katz**         **Yonatan Belinkov**
Technion - Israel Institute of Technology
shachar.katz@cs.technion.ac.il         belinkov@technion.ac.il

## Abstract

Recent advances in interpretability suggest we can project weights and hidden states of transformer-based language models (LMs) to their vocabulary, a transformation that makes them more human interpretable. In this paper, we investigate LM attention heads and memory values, the vectors the models dynamically create and recall while processing a given input. By analyzing the tokens they represent through this projection, we identify patterns in the information flow inside the attention mechanism. Based on our discoveries, we create a tool to visualize a forward pass of Generative Pre-trained Transformers (GPTs) as an interactive flow graph, with nodes representing neurons or hidden states and edges representing the interactions between them. Our visualization simplifies huge amounts of data into easy-to-read plots that can reflect the models' internal processing, uncovering the contribution of each component to the models' final prediction. Our visualization also unveils new insights about the role of layer norms as semantic filters that influence the models' output, and about neurons that are always activated during forward passes and act as regularization vectors. [1]

## 1   Introduction

*Wouldn't it be useful to have something similar to an X-ray for transformers language models?*

Recent work in interpretability found that hidden-states (HSs), intermediate activations in a neural network, can reflect the "thought" process of transformer language models by projecting them to the vocabulary space using the same transformation that is applied to the model's final HS, a method known as the "logit lens" (nostalgebraist, 2020). For instance, the work of Geva et al. (2021, 2022b) shows how the fully-connected blocks of

---

[1]Code and tool are available at https://github.com/shacharKZ/VISIT-Visualizing-Transformers.

transformer LMs add information to the model's residual stream, the backbone route of information, promoting tokens that eventually make it to the final predictions. Subsequent work by Dar et al. (2022) shows that projections of activated neurons, the static weights of the models' matrices, are correlated in their meaning to the projections of their block's outputs. This line of work suggests we can stop reading vectors (HSs or neurons) as just numbers; rather, we can read them as words, to better understand what models "think" before making a prediction. These studies mostly interpret static components of the models or are limited to specific case studies that require resources or expertise.

To address the gap in accessibility of the mechanisms behind transformers, some studies create tools to examine how LMs operate, mostly by plotting tables of data on the most activated weights across generations or via plots that show the effect of the input or specific weights on a generation (Geva et al., 2022a; Hoover et al., 2020). Yet, such tools do not present the role of each of the LM's components to get the full picture of the process.

In this paper, we analyze another type of LMs' components via the logit lens: the attention module's dynamic memory (Vaswani et al., 2017), the values (HS) the module recalls from previous inputs. We describe the semantic information flow inside the attention module, from input through keys and values to attention output, discovering patterns by which notions are passed between the LM's components into its final prediction.

Based on our discoveries, we model GPTs as flow-graphs and create a dynamic tool showing the information flow in these models (for example, Figure 1). The graphs simplify detection of the effect that single to small sets of neurons have on the prediction during forward passes. We use this tool to analyze GPT-2 (Radford et al., 2019) in three case studies: (1) we reflect the mechanistic analysis of Wang et al. (2022) on indirect object identification

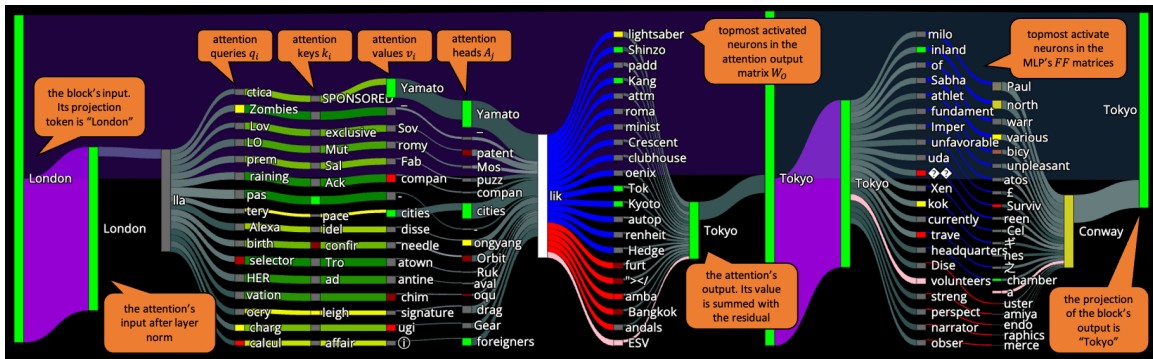

Figure 1: Modeling a single layer (number 14) of GPT-2 for the prompt: "The capital of Japan is the city of". Each node represents a small group of neurons or HS, which are labeled by the top token of their projection to the vocabulary space. The plot should be read from left to right and includes the attention block: LN (the node at the end of the first purple edge), query, memory keys and values (with greenish edges) and the topmost activated attention output neurons (in blue and red), followed by the MLP: LN (in purple), first and second matrices' most activated neurons (in blue and red). The dark edges in the upper parts of the plot are the residuals of each sub-block.

in a simple way; (2) we analyze the role of layer norm layers, finding they act as semantic filters; and (3) we discover neurons that are always activated, related to but distinct from rogue dimensions (Timkey and van Schijndel, 2021), which we term regularization neurons.

## 2 Background

### 2.1 The Transformer Architecture

We briefly describe the computation in an autoregressive transformer LM with multi-head attention, such as GPT-2, and refer to Elhage et al. (2021) for more information.[2][3]

The model consists of a chain of blocks (layers), that read from and write to the same residual stream. The input to the model is a sequence of word embeddings, $x_1, \ldots, x_t$ (the length of the input is $t$ tokens), and the residual stream propagates those embeddings into deeper layers, referring to the intermediate value it holds at layer $l$ while processing the $i$-th token as $hs_i^l$ ($hs_i$ in short). The HS at the final token and top layer, $hs_t^L$, is passed through a layer norm, $ln_f$, followed by a decoding matrix $D$ that projects it to a vector the size of the vocabulary. The next token probability distribution is obtained by applying a softmax to this vector.

Each block is made of an attention sub-block (module) followed by a multi-layer perceptron (MLP), which we describe next.

---

[2]Appendix A details these models in the context of our graph modeling.

[3]For simplicity we do not mention dropout layers and position embeddings here.

### 2.2 GPTs Sub-Blocks

**Attention:** The attention module consists of four matrices, $W_Q, W_K, W_V, W_O \in \mathbb{R}^{d \times d}$. Given a sequence of HS inputs, $hs_1, \ldots, hs_t$, it first creates three HS for each $hs_i$: $q_i = hs_i W_Q$, $k_i = hs_i W_k$, $v_i = hs_i W_v$, referred to as the current queries, keys, and values respectively. When processing the $t$-th input, this module stacks the previous $k_i$'s and $v_i$'s into matrices $K, V \in \mathbb{R}^{d \times t}$, and calculates the attention score using its current query $q = q_t$:
$$A = Attention(q, K, V) = softmax(\frac{qK^\top}{\sqrt{d}})V.$$
In practice, this process is done after each of $q_i, k_i, v_i$ is split into $h$ equal vectors to run this process in parallel $h$ times (changing the dimension from $d$ to $d/h$) and to produce $A_j \in \mathbb{R}^{\frac{d}{h}}$ ($0 \le j < h$), called heads. To reconstruct an output in the size of the embedding space, $d$, these vectors are concatenated together and projected by the output matrix: $Concat(A_0, ..., A_{h-1})W_O$. We refer to the process of this sub-block as $Attn(hs)$.

We emphasize that this module represents dynamic memory: it recalls the previous values $v_i$ (which are temporary representations for previous inputs it saw) and adds a weighted sum of them according to scores it calculates from the multiplication of the current query $q_t$ with each of the previous keys $k_i$ (the previous keys and values are also referred to as the "attention k-v cache").

**MLP:** This module consists of an activation function $f$ and two fully connected matrices, $FF_1, FF_2^\top \in \mathbb{R}^{d \times N}$ ($N$ is a hidden dimension, usually several times greater than $d$). Its output is $MLP(x) = f(xFF_1)FF_2$.

**Entire block:** GPT-2 applies layer norm (LN), before each sub-block: $ln_1$ for the attention and $ln_2$ for the MLP. While LN is thought to improve numeric stability (Ba et al., 2016), one of our discoveries is the semantic role it plays in the model (subsection 5.2). The output of the transformer block at layer $l$, given the input $hs_i^l$, is

$$hs_i^{l+1} = hs_i^l + Attn(ln_1(hs_i^l)) + \\ MLP(ln_2(Attn(ln_1(hs_i^l)) + (hs_i^l))) \quad (1)$$

### 2.3 Projecting Hidden States and Neurons

**The Logit Lens (LL):** nostalgebraist (2020) observed that, since the decoding matrix in GPTs is tied to the embedding matrix, $D = E^\top$, we can examine HS from the model throughout its computation. Explicitly, any vector $x \in \mathbb{R}^d$ can be interpreted as a probability on the model's vocabulary by projecting it using the decoding matrix with its attached LN:

$$LL(x) = softmax(ln_f(x)D) = s \in \mathbb{R}^{|vocabulary|} \quad (2)$$

By applying the logit lens to HS between blocks, we can analyze the immediate predictions held by the model at each layer. This allows us to observe the incremental construction of the model's final prediction, which Geva et al. (2022b) explored for the MLP layers.

Very recent studies try to improve the logit lens method with additional learned transformations (Belrose et al., 2023; Din et al., 2023). We stick with the basic approach of logit lens since we wish to explore the interim hypotheses formed by the model, rather than better match the final layer's output or shortcut the model's computation, and also, since those new methods can only be applied to the HS between layers and not to lower levels of components like we explain in the next section.

**Interpreting Static Neurons:** Each of the mentioned matrices in the transformer model shares one dimension (at least) with the size of the embedding space $d$, meaning we can disassemble them into neurons, vectors that correspond to the "rows" or "columns" of weights that are multiplied with the input vector, and interpret them as we do to HS. Geva et al. (2021) did this with single neurons in the MLP matrices and Dar et al. (2022) did this with the interaction of two matrices in the attention block, $W_Q$ with $W_K$ and $W_V$ with $W_O$, known as the transformer circuits $QK$ and $OV$ (Elhage et al., 2021). [4] These studies claim that activating a neuron whose projection to the vocabulary has a specific meaning (the common notion of its most probable tokens) is associated with adding its meaning to the model's intermediate processing.

In our work we interpret single and small groups of HS using the logit lens, specifying when we are using an interaction circuit to do so. In addition, while previous studies interpret static weights or solely the attention output, we focus on the HS that the attention memory recalls dynamically.

## 3 Tracing the Semantics Behind the Attention's Output

In this section, we trace the components which create the semantics of the attention block's output, by comparing vectors at different places along the computation graph. In all the following experiments, we project HS into the vocabulary using the logit lens to get a ranking of all the tokens, then pick the top-$k$ tokens according to their ranking. We measure the common top tokens of two vectors ($x_1$ and $x_2$) via their intersection score $I_k$ (Dar et al., 2022):

$$I_k(x_1, x_2) = \frac{LL(x_1)[\text{top-k}] \cap LL(x_2)[\text{top-k}]}{k} \quad (3)$$

We say that two vectors are semantically aligned if their $I_k$ is relatively high (close to 1) since it means that a large portion of their most probable projected tokens is the same.

Throughout this section, we used CounterFact (Meng et al., 2022), a dataset that contains factual statements, such as the prompt *"The capital of Norway is"* and the correct answer *"Oslo"*. We generate 100 prompts randomly selected from CounterFact using GPT-2-medium, which we verify the model answers correctly. We collect the HSs from the model's last forward-passes (the passes that plot the answers) and calculate $I_{k=50}$. [5]

### 3.1 Projecting the Attention Memory

For our analysis we interpret $W_V$ products, the attention's heads $A_j$ and its memory values, $v_{ji}$ ($j$ for head index and $i$ for token index). For each component we calculate its mean $I_{k=50}$ with its attention block output ($Attn(hs_i^l)$, "$I_k$ attn"), its

---

[4] To achieve two matrix circuit we multiply one matrix with the output of the second, for example, the $OV$ circuit outputs are the multiplication of $W_O$ matrix with the outputs of $W_V$.

[5] Refer to Appendix C, D.1 for more information about our model selection and setup.

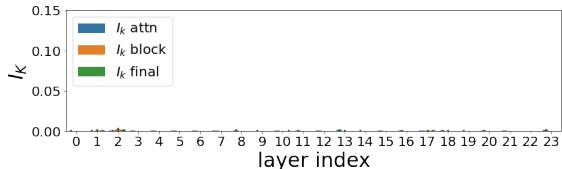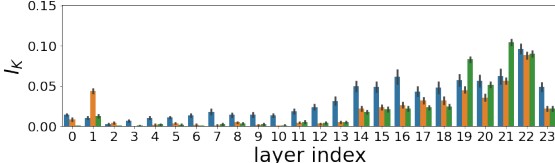

| (a) Without $W_O$ projection | (b) With $W_O$ projection |

Figure 2: Comparing $I_{k=50}$ token projection alignment between the mean of all heads $A_j$ with different parts of the model's, with and without using the attention block's $W_O$ matrix for projection, suggesting that the output of the attention block's $W_V$ operates in a different space and that $W_O$'s role is to adjust it to the common embedded space.

transformer block output ($hs_i^{l+1}$, "$I_k$ block"), and the model's final output ($hs_i^L$, "$I_k$ final").

Dar et al. (2022) suggest using the $OV$ circuit, in accordance to Elhage et al. (2021), to project the neurons of $W_V$ by multiplying them with $W_O$. Similarly, we apply logit lens to $A_j$ once directly and once with the $OV$ circuit, by first multiplying each $A_j$ with the corresponding parts of $W_O$ to the $j$-th head ($j : j + \frac{d}{h}$). [6] While the first approach shows no correlation with any of the $I_k$ we calculate (Figure 2a), the projection with $OV$ shows semantic alignment that increase with deeper layers, having some drop at the final ones (Figure 2b). The pattern of the latter is aligned with previous studies that examine similar scores with the MLP and the entire transformer block (Haviv et al., 2023; Lamparth and Reuel, 2023; Geva et al., 2022b), showing that through the $OV$ circuit there is indeed a semantic alignment between the attention heads and the model's outputs and immediate predictions.

This finding suggests that the HS between $W_V$ and $W_O$ do not operate in the same embedded space, but are rather used as coefficients of the neurons of $W_O$. Therefore, outputs of $W_V$ should be projected with logit lens only after they are multiplied by $W_O$.

### 3.2 Projecting Only the Top Attention Heads

We observe that at each attention block the norms of the different heads vary across generations, making the top tokens of the heads with the largest norms more dominant when they are concatenated together into one vector. Therefore, we separately ranked each attention block's heads with the $OV$ circuit ($A_j W_O$) according to their norms and repeated the comparison. We found that only the

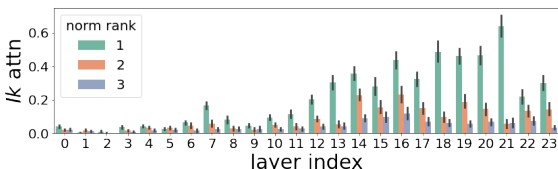

(a) Mean $I_{k=50}$ for only the top 3 heads with the largest norm, comparing to attention block output.

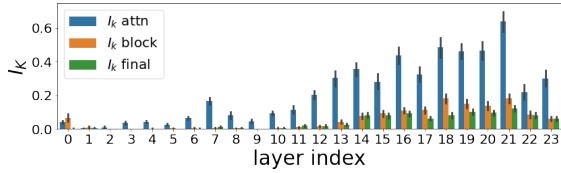

(b) Mean $I_{k=50}$ for only the head with the largest norm, comparing to attention block output, layer output and the model's final output.

Figure 3: Projecting attention heads

few heads with the largest norm have a common vocabulary with their attention block output (Figure 3a), which gradually increases the effect on the blocks' outputs and the final prediction (Figure 3b). This suggests that the attention block operates as a selective association gate: by making some of the heads much more dominant than others, this gate chooses which heads' semantics to promote into the residual (and which to suppress).

### 3.3 Projecting Memory Values

We ran the same experiment comparing the memory values $v_{ji}$, the values that the attention mechanism recalls from the previous tokens. For each head $A_j$, we rank its memory values based on their attention scores and observe that memory values assigned higher attention scores also exhibit a greater degree of semantic similarity with their corresponding head. The results for the top three memory values are illustrated in Figure 5.

---

[6]In practice, the implementation of projecting a vector in the size of $\frac{d}{h}$ like $A_j$ is done by placing it in a $d$-size zeroed vector (starting at the $j \cdot \frac{d}{h}$ index). Now we can project it using logit lens (with or without multiplying it with the entire $W_O$ matrix for the $OV$ circuit).

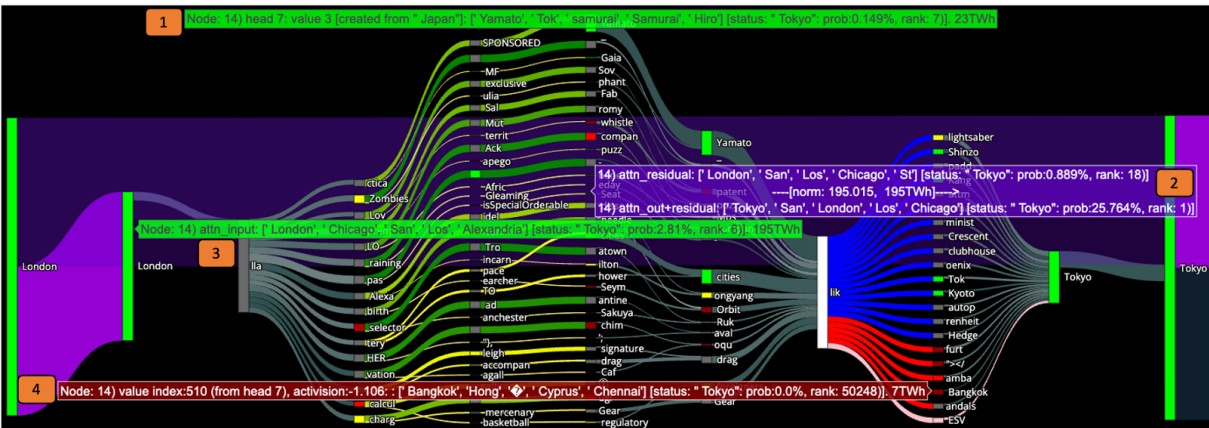

Figure 4: Modeling a single attention block of GPT-2 for the prompt: "The capital of Japan is the city of". The pop-up text windows are (from top to bottom): One of the memory values, whose source is the input token "Japan" and whose projection is highly correlated with the output of the model, "Tokyo" (1). The residual stream and the labels of its connected nodes (2). The input to the attention block after normalization, which its most probable token is "London" (3). One of the most activated neurons of $W_O$ that has a negative coefficient. Its projection is highly unaligned with the model's output, which the negative coefficient suppresses (4). At the block's input, the chance for "Tokyo" is $< 1\%$, but at its output it is $25\%$ (purple pop-up window (2)), i.e., this attention block prompts the meaning of "Tokyo". The two biggest heads are "Yamato" (with Japanese concepts) and "cities", which together create the output "Tokyo".

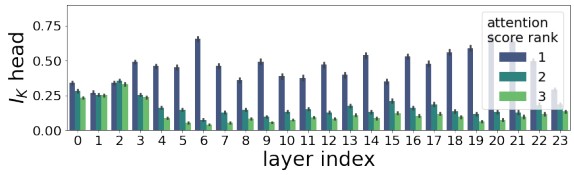

Figure 5: Mean $I_{k=50}$ for the 3 top biggest by attention score memory values, comparing to their head output.

### 3.4 Interim Summary

The analysis pictures a clear information flow, from a semantic perspective, in the attention block: [1] the block's input creates a distribution on the previous keys resulting in a set of attention scores for each head (subsection 2.2), [2] which trigger the memory values created by previous tokens, where only the ones with the highest attention scores capture the head semantics (subsection 3.3). [3] The heads are concatenated into one vector, promoting the semantics of only a few heads (subsection 3.2) after they are projected to the vocabulary through $W_O$ (subsection 3.1). An example of this procedure is shown for the prompt "The capital of Japan is the city of", with the expected completion "Tokyo", in Figure 1 for the flow in a full block and in Figure 4 for the flow in the attention sub-block. An input token like "Japan" might create a memory value with the meaning of Japanese concepts, like "Yamato" and "Samurai". This memory value can capture its head meaning. Another head might have the meaning of the token "cities", and together the output of the attention could be "Tokyo".

## 4 Modeling the Information Flow as a Flow-Graph

As in most neural networks, information processing in an autoregressive LM can be viewed as a flow graph. The input is a single sentence (a sequence of tokens), the final output is the probability of the next word, with intermediate nodes and edges. Geva et al. (2022b, 2021) focused on information flow in and across the MLP blocks, while our analysis in section 3 focused on the information flow in the attention block. In this section, we describe how to construct a readable and succinct graph for the full network, down to the level of individual neurons. Our graph is built on collected HS from a single forward pass: it uses single and small sets of HSs as nodes, while edges are the interactions between nodes during the forward pass.

One option for constructing a flow graph is to follow the network's full computation graph. Common tools do this at the scale of matrices (Roeder, 2017), coarser than the neuronal scale we seek. They usually produce huge, almost unreadable, graphs that lack information on which values are passed between matrices and their effect. Similarly, if we were to connect all possible nodes

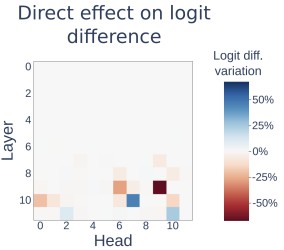

(a) Tabular information from Wang et al. (2022) for GPT-2 small, identifying the Name Mover and Negative Name Mover Heads, measured by strongest direct effect on the final logits.

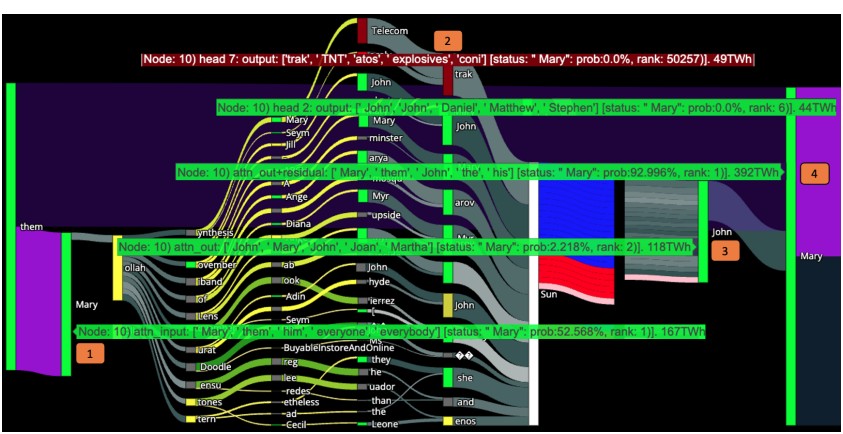

(b) The corresponding flow-graph for layer 10 attention from GPT-2 small.

Figure 6: While 6a shows quantitative results, the graph model in Figure 6b shows qualitative information that is otherwise difficult to notice: The attention's LN is the first place in the model where the attention input's most probable token is " Mary" (1). The Negative Name Mover Head from layer 10, represented by the blue cell in the table's 10-th row, is visualized in the graph with a red pop-up showing it assigns the token " Mary" its lowest possible ranking, meaning its role is to reduce the probability of this token (2). The output of the attention block is the token " John" but its second most probable output is " Mary" with around 2% chance (3). However, when added to the residual, together they predict almost 93% chance for " Mary" (4).

(neurons and HSs) and edges (vector multiplications and summation), the graph would be unreadable, as there are thousands of neurons in each layer. Moreover, our analysis in section 3 shows that many components are redundant and do not affect the model's intermediate processing. Therefore, based on the hypothesis that the neurons with the strongest activations exert more significant influences on the output, we prune the graph to retain only the most relevant components: by assigning scores to the edges at each computation step, like ranking the attention scores for the edges connected to each memory value, or the activation score for neurons in MLPs, we present only the edges with the highest scores at each level. Nodes without any remaining edge are removed. The goal is to present only the main components that operate at each block. See subsection A.4 for details.

To present the semantic information flow, we assign each node with its most probable projected token and the ranking it gives to the model's final prediction, according to the logit lens. Each node is colored based on its ranking, thereby emphasizing the correlation between the node's meaning and the final prediction. Additionally, we utilize the width of the edges to reflect the scores used for pruning.

Figures 1 and 4 show static examples on one sentence, the first for a single transformer block's graph and the second with an annotated explanation on the attention sub-blocks's sub-graph.

## 5 Example of Use and Immediate Discoveries

The flow-graph model is especially beneficial for qualitative examinations of LMs to enhance research and make new discoveries. In this section, we demonstrate this with several case studies.

### 5.1 Indirect Object Identification

Recently, Wang et al. (2022) tried to reverse-engineer GPT-2 small's computation in indirect object identification (IOI). By processing prompts like "When Mary and John went to the store, John gave a drink to", which GPT-2 small completes with "Mary", they identified the roles of each attention head in the process using methods like changing the weights of the model to see how they affect its output. One of their main discoveries was attention heads they called Name Mover Heads and Negative Name Mover Heads, due to their part in copying the names of the indirect object (IO, "Mary") or reducing its final score.

We ran the same prompt with the same LM and examined the flow-graph it produced. The flow graph (Figure 6b) is highly correlated to Wang et al.'s results (Figure 6a). While they provide a table detailing the impact of each attention head on the final prediction, our graph shows this by indicating which token each head promotes. For instance, heads that project the token "Mary" among their most probable tokens are the Name Mover Heads,

while Negative Name Mover heads introduce the negative meaning of "Mary" (evident by the low probability of "Mary" in their projection, highlighted in red). Not only does our model present the same information as the paper's table, which was produced using more complex techniques, but our modeling also allows us to observe how the attention mechanism scores each previous token and recalls their memory values. For example, we observe that the Negative Name Mover in layer 10 obtains its semantics from the memory value produced by the input token "Mary".

We do not claim that our model can replace the empirical results of Wang et al. (2022), but it could help speed up similar research processes due to the ability to spot qualitative information in an intuitive way. Also, the alignment between the two studies affirms the validity of our approach for a semantic analysis of information flow of GPTs.

## 5.2 Layer Norm as Sub-Block Filter

Layer norm (LN) is commonly applied to sub-blocks for numerical stability (Ba et al., 2016) and is not associated with the generation components, despite having learnable weights. We investigate the role of LN, focusing on the first LN inside a GPT-2 transformer block, $ln_1$, and apply the logit lens before and after it. We use the data from section 3 and, as a control group, random vectors. Figure 7 shows change in logit lens probability of all tokens after applying LN. The tokens whose probability decreases the most are function words like "the", "a" or "not", which are also tokens with high mean probability across our generations (although they are not the final prediction in the sampled generations). Conversely, tokens that gain most probability from LN are content words like "Microsoft" or "subsidiaries". See more examples and analyses of the pre-MLP LN, $ln_2$, in Appendix E. These results suggest that the model uses LN to introduce new tokens into the top tokens that it compares at each block.

## 5.3 Regularization Neurons

While browsing through many examples with our flow graph model, we observed some neurons that are always activated in the MLP second matrix, $FF_2$. We quantitatively verified this using data from section 3 and found that each of the last layers (18—23) has at least one neuron that is among the 100 most activated neurons more than $85\%$ of the time (that is, at the top $98\%$ most activated

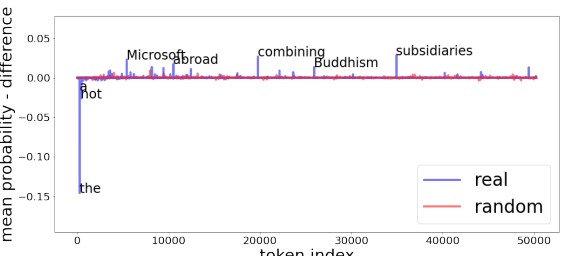

Figure 7: Differences in token probabilities before and after LN $ln_1$ from layer 15 of GPT-2 medium, according to the generations from section 3. The horizontal axis is the index of all the tokens in GPT-2 and the vertical shows if the token lost or gained probability from the process (negative or positive value). We annotate the tokens that are most affected.

neurons out of 4096 neurons in a given layer). At least one of these neurons in each layer results in function words when projected with the logit lens, which are invalid generations in our setup. We further observe that these neurons have exceptionally high norms, but higher-entropy token distributions (closer to uniform), when projected via the logit lens (Figure 8). This suggests that these neurons do not dramatically change the probabilities of the final predictions.

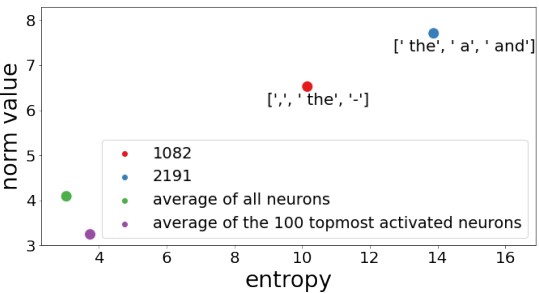

Figure 8: Entropy and norm of "regularization neurons" from the second MLP matrix of layer 19 compared to the matrix average and the 100 most activated neurons across 100 prompts from CounterFact.

By plotting these neurons' weights, we find a few outlier weights with exceptionally large values (Figure 9). Since these neurons are highly activated, the outlier weights contribute to the phenomenon of outlier or rogue dimensions in the following HS, described in previous work (Puccetti et al., 2022; Timkey and van Schijndel, 2021; Kovaleva et al., 2021). This line of work also shows that ignoring those dimensions can improve similarity measures between embedded representations, while ignoring them during the computation of the model causes a

significant drop in performance.

Our analysis adds a semantic perspective to the discussion on rogue dimensions: since these neurons' projections represent "general" notions (not about a specific topic, like capitals or sports) and since they have high entropy, they might play a role of regularization or a sort of bias that is added as a constant to the residual stream. Finally, to reflect such cases, we paint all the accumulation edges in our flow-graph (where vectors are summed up) in grey, with darker shades expressing lower entropy.

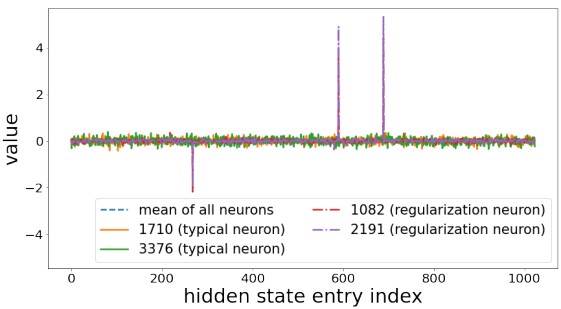

Figure 9: Plotting the value in each entry in the regularization neurons at layer 19, comparing the mean neuron and presenting two randomly sampled neurons that represent typical neurons. Those high magnitudes of the 3 entries in the regularization neurons help in the creation of the rogue dimensions phenomena.

## 6 Related Work

Derived from the original logit lens (nostalgebraist, 2020), several studies analyze the role of each component in LMs using token projection (Geva et al., 2022b; Dar et al., 2022). In the last few months, new studies suggest trainable transformation for projecting HS (Din et al., 2023; Belrose et al., 2023), promising to better project HS in the earlier layers of LMs (which currently seems to have less alignment with the final output than later ones).

Other work took a more mechanistic approach in identifying the role of different weights, mostly by removing weights or changing either weights or activations, and examining how the final prediction of the altered model is affected (Wang et al., 2022; Meng et al., 2022, 2023; Dai et al., 2022).

There has been much work analyzing the attention mechanism from various perspectives, like trying to assign linguistic meaning to attention scores, questioning their role as explanations or quantify its flow (Abnar and Zuidema, 2020; Ethayarajh and Jurafsky, 2021). See Rogers et al. (2020) for an overview.

Our work is different from feature attribution methods (Ribeiro et al., 2016; Lundberg and Lee, 2017), which focus on identifying the tokens in the input that exert a greater influence on the model's prediction. Some studies visualise the inner computation in LMs. For example, the work of Geva et al. (2022a) tries to look into the inner representation of model by visualizing the logit lens projection of the HSs between blocks and on the MLP weights. Other tools that focused on the attention described the connection between input tokens (Hoover et al., 2020; Vig and Belinkov, 2019) but did not explore the internals of the attention module. There are general tools for visualizing deep learning models, like Roeder (2017), but they only describe the flow of information between matrices, not between neurons. Strobelt et al. (2018a,b) visualize hidden states and attention in recurrent neural network models, allowing for interaction and counterfactual exploration.

## 7 Conclusion

In this work, we used token projection methods to trace the information flow in transformer-based LMs. We have analyzed in detail the computation in the attention module from the perspective of intermediate semantics the model processes, and assessed the interactions between the attention memory values and attention output, and their effect on the residual stream and final output.

Based on the insights resulting from our analysis, we created a new tool for visualizing this information flow in LMs. We conducted several case studies for the usability of our new tool, for instance revealing new insights about the role of the layer norm. We also confirmed the validity of our approach and showed how it can easily support other kinds of analyses.

Our tool and code will be made publicly available, in hope to support similar interpretations of various auto-regressive transformer models.

### Limitations

Our work and tool are limited to English LMs, in particular different types of GPT models with multi-head attention, and the quantitative analyses are done on a dataset of factual statements used in recent work. While our methodology is not specific to this setting, the insights might not generalize to

other languages or datasets.

In this work we interpret HS and neurons using projection methods which are still being examined, as well the idea of semantic flow. The way we measure impact and distance between HS using $I_k$ (the intersection between their top tokens) is not ideal since it might not convey the semantic connection of two different tokens with the same meaning. While it is possible to achieve more nuanced measurements with additional human resources (users) or semi-automatic techniques, there would be limitations in mapping a vast number of neurons and their interactions due to the enormous number of possible combinations. Therefore, we deliberately chose not to employ human annotators in our research.

Our pruning approach is based on the assumption that the most activate neurons are the ones that determine the model's final prediction. Although this claim is supported by our qualitative analysis, we cannot claim that the less activated neurons are not relevant for building the prediction. Since our flow-graph model does not show those less active neurons, it might give misleading conclusions.

Finally, our methods do not employ causal techniques, and future work may apply various interventions to verify our findings. Our tool tries to reflect what GPT "thinks", but further investigation of its mechanism is needed before approaching a full understanding of this "black box".

## Acknowledgements

This work was supported by the ISRAEL SCIENCE FOUNDATION (grant No. 448/20), Open Philanthropy, and an Azrieli Foundation Early Career Faculty Fellowship.

## Ethics Statement

Our goal is to improve the understanding of LMs by dissecting inner layers and intermediate results of GPT. The semantics behind some projections might appear offensive and we want to be clear that we have no intention of such. Further work might use our new tool to try to identify components of the model that control a given idea or knowledge, and to edit it. We hope such a use case would be for better representing information and not for spreading any hate.

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

## A   Modeling GPTs as a Flow-Graph

This section presents a formal construction of GPTs as flow-graphs for single forward passes, followed by more implementation details. The information here supplements the brief description given in sub-section 2.2 and is brought here for completeness.

Like any graph, our graph is defined by a set of nodes (vertices) and edges (links). In our case, the graph follows a hierarchical structure, starting with the breakdown of the entire model into layers, followed by sub-blocks such as attention and MLP blocks, and eventually individual or small sets of neurons. A GPT model consisting of $L$ transformer blocks denoted as $B_l$ ($0 \leq l < L$), where $W_Q$, $W_K$, $W_V$, $W_O$ represent the matrices for the attention block, and $FF_1 = W_{FF1}$ and $FF_2 = W_{FF2}$ represent the matrices for the MLP. We now walk through the forward computation in the model and explain how we construct the flow graph. Figure 10 over-viewing the process.

### A.1   The Attention Block as a Flow-Graph

1. The input to the $l$-th block for the $t$-th input, $hs_t^l$, passes through a LN, resulting in a normalized version of it. We create a node for the input vector and another node for the normalized vector, connecting them with an edge.

2. The normalized input is multiplied by $W_Q$, $W_K$, and $W_V$, resulting in query, key, and value representations ($q$, $k$, $v$). We create a single node to represent these three representations, as they are intermediate representations used by the model. We construct an edge between the normalized input and this node.

3. The last three representations ($q$, $k$, $v$) are split into $h$ heads ($q_{jt}$, $k_{jt}$, $v_{jt}$ for $0 \leq j < h$). Each head's query vector ($q_{jt}$) is multiplied by all the previous key vectors ($k_{ji}$ for $1 \leq i \leq t$), calculating the attention probability for each of the previous token values. We create a node for each head's query vector and connect it with an edge to the overall query node created in the previous step. Additionally, we create a node for each key vector and connect it with an edge to its corresponding head's query vector.

4. Each memory value vector ($v_{ji}$, the memory value of the $j$-th head and the $i$-th input token), is summed up with a coefficient (the attention score) into its corresponding head $A_j$. We create a node for each value vector and connect it with an edge to its corresponding key vector. Furthermore, we create a node for each summed-up head $A_j$ and connect it to all of its memory value vectors. This establishes a direct path between each head's query $q_{jt}$, its keys $k_{ji}$, its values $v_{ji}$, and the head's final vector $A_j$. It is important to note that the calculation of attention scores is non-linear and preserves the relative ranking among memory values.

5. The $h$ heads $A_j$ are concatenated, resulting in a vector $A_{concatenated}$ with the same size as the model's hidden state (embedding size). We create a node for $A_{concatenated}$ and connect all the heads $A_j$ to it.

6. $A_{concatenated}$ is multiplied by $W_O$ to produce the attention output $Attn(hs_t^l)$. We create a node for each entry in $A_{concatenated}$ and each neuron in $W_O$, connecting them through edges representing the multiplication process. Additionally, we create a node for the output $Attn(hs_t^l)$ and connect each neuron to it.

7. The attention output is then added to the residual stream of the model. We create a node for the sum of the attention block and the residual, $hs_{attn+residual}$, and connect it to $Attn(hs_t^l)$.

8. The attention block also contains a skip connection, The residual, from the input $hs_t^l$ straight to the output $hs_{attn+residual}$, so we connect an edge between them.

### A.2   The Feed Forward Block as a Flow-Graph

This structure is mainly based on the theory of using two fully connected layers as keys and values, as described by Geva et al. (2021)

1. Similar to the attention block, the input to this block, denoted as $\hat{hs}_l^t = hs_{attn+residual}$ (representing the intermediate value of the residual after the attention sub-block), passes through a layer norm, resulting in a normalized version of it. We create a node for the input vector and another node for the normalized vector, connecting them with an edge.

2. The normalized input is multiplied by $W_{FF1}$. For each neuron in the matrix, we create a

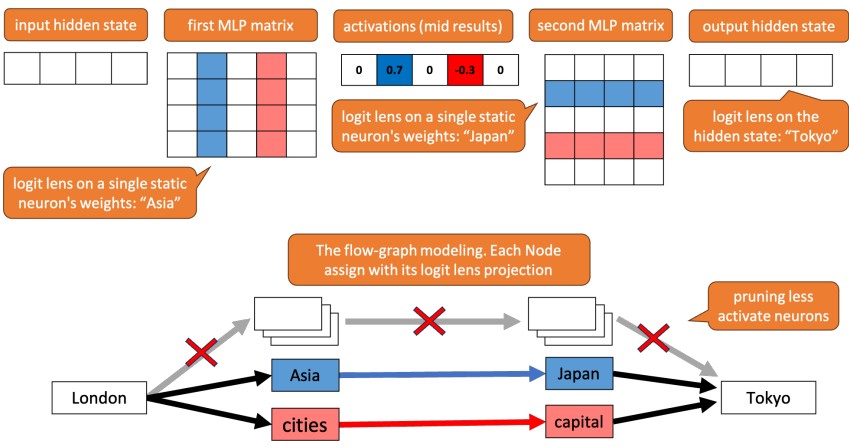

Figure 10: The overview process of creating a flow-graph modeling (the bottom graph) from a single forward pass (the upper draw). In this toy-example we model a simplified version of a MLP sub-block. Each node in the graph is correspond to a static weight or HS in the upper diagram and labeled by its logic lens projection, for example: the input has the meaning of "London" and the output has the meaning of "Tokyo".

node and connect an edge from the normalized input to it (corresponding to the multiplication of each neuron separately).

3. The result of the previous multiplication is a vector of coefficients for the second MLP matrix, $W_{FF2}$. Consequently, we create a node for each neuron in $W_{FF2}$ and connect an edge between each neuron and its corresponding neuron from $W_{FF1}$. It is important to note that the actual process includes a non-linear activation between the two matrices, which affects the magnitude of each coefficient but not its sign (positive or negative).

4. The neurons of $W_{FF2}$ are multiplied by their coefficients and summed up into a single vector, which serves as the output of the MLP block, denoted as $MLP(\hat{hs}_l^t)$. We create a node for $MLP(\hat{hs}_l^t)$ and connect all the neurons from $W_{FF2}$ to it.

5. The output of the block is then added to the model's residual stream. We create a node for the sum of the MLP block and the residual, denoted as $hs_{MLP+residual}$, and connect it to $MLP(\hat{hs}_l^t)$.

6. Similarly to the attention block, the MLP block also includes a skip connection, directly connecting the input $\hat{hs}_l^t$ to the output $hs_{MLP+residual}$. Therefore, we connect an edge between them.

### A.3 Connecting The Graphs of Single Blocks Into One

In GPT-2 each transformer block contains an attention block followed by a MLP block. We define a graph for each transformer block by the concatenation of its attention graph and MLP graph, where the two graphs are connected by an edge between the attention's $hs_{attn+residual}$ and the MLP's $\hat{hs}_l^t$. The input to the new graph is the input of the original attention sub-graph, and its output is the output of the original MLP sub-graph.

To define the graph of the entire model we connect all its transformer blocks' sub-graphs into one graph by connecting an edge between each block's sub-graph output and the input of its following block's sub-graph. The input to the new graph is the input of the first block and the output is the final block output.

### A.4 Scoring the Nodes and Edges

In order to emphasize some of the behaviors of the models, we define scoring functions for its nodes and edges.

**Scoring nodes according to projected token ranking and probability:** as we described, each node is created from a vector that we project to the vocabulary space, resulting in a probability score that defines the ranking of all the model's tokens. Given a specific token $w$ and a single vector $v$ we define its neuronal ranking and probability, $v_{rank}(w)$ and $v_{prob}(w)$, as the index and probability of token $w$ in the projected vector of $v$.

**Scoring edges according to activation value and norm:** There are two types of edges: edges that represent the multiplication of neurons with coefficients (representing neuron activation) and edges that represent summation (as part of matrix multiplication). Edges that represent multiplication with coefficients are scored by the coefficient. We also include in this case the attention scores, which are

used as coefficients for the memory values. Edges that represent summation are scored by the norm of the vector which they represent. This scoring aims to reflect the relative involvement of each of the weights, since previous work found that neurons with higher activation or norm have a stronger impact on the model behavior (Geva et al., 2022b).

## A.5 Modeling a Single GPT Inference as a Flow-Graph

Given a prompt $x_1, \ldots, x_t$ we pass it through a GPT model and collect every HS (input and output of each matrix multiplication). Then we create the flow-graph as described above, where the input and HS are according to the last input token $x_t$ and the attention memory (previous keys and values) correspond to all the input tokens. This process results in a huge graph with many thousands of nodes even for small models like GPT-2 small (Radford et al., 2019), which in this sense can only be examined as tabular data, similar to previous work. Since our goal is to emphasize the flow of data, we reduced the number of nodes according to our discoveries and the assumption that neurons in the MLP blocks with relatively low activation have a small effect on the model output (Geva et al., 2022b). We also note that with a simple adjustment our model can show any number of neurons or show only chosen ones.

The reduced graph is defined as follows:

- In the attention sub-graph, we chose to present all the nodes of the heads' query and output, $q_{jt}, A_j$, but to present only the memory keys and values, $k_{ji}, v_{ji}$, that received the highest attention score, in light of the results from Section 3.1. We also decided to present only the top most activated neurons of $W_O$, according to the largest entries (by absolute value) from its coefficients HS, $A_{concated}$.

- In the MLP sub-graph we decided to show only the nodes of the most activated neurons. The activation is determined by the highest absolute values in the HS between the two matrices after the nonlinearity activation. That is, we examine the input to the second MLP matrix $W_{FF2}$ and present only the nodes that are connected to its highest and lowest entries.

- We make it possible to create a graph from only part of consecutive transformer blocks, allowing us to examine only a few blocks at a time.

The above simplifications help construct a scalable graph that humans can easily examine.

## A.6 Implementation Details and how to Read the Graph

We use the Python package Plotly-Express (Inc., 2015) to create a plot of the model. We will provide all the source code we created to model the GPT-2 family models (small, medium, large and XL) and GPT-J (Wang and Komatsuzaki, 2021), which includes configuration files that allow adjusting the tool to other decoders with multi-head attention. We are also providing the code to be used as a guided example with instructions designed to facilitate the adaptation of our flow-graph model to other GPT models.

Using our tool is straightforward and only requires running our code. The flow-graph plot can be presented in your software environment or saved as an HTML file to view via a browser. Personal computers and environments like Google Colab are sufficient for modeling LMs like GPT-2 medium, even without GPU. Plotly-Express allows us to inspect the created graphs interactively, like seeing additional information when hovering over the nodes and edges, or to filter some of them by the "Select" options on the top right of the generated plots.

The basics on how to read and use the flow-graph plots are:

- The flow is presented from left to right (matrices that operate earlier during the forward pass will be to the left of later ones). When plotting a single block we can identify the attention sub-block (the first from the left) and the MLP sub-block as they are connected by a wide node and by separate and parallel wide edges representing the residual (each with a slightly different color). When plotting more than one block we can identify the different blocks by the repetitive structure of each.

- Each node is labeled with its most probable projected token. When hovering over a node, we can see from which layer and from which HS or matrix it was taken (the first number and the follow-up text in the pop-up text window. For example: "10) attn-input" suggest this node is the input of the attention sub-block

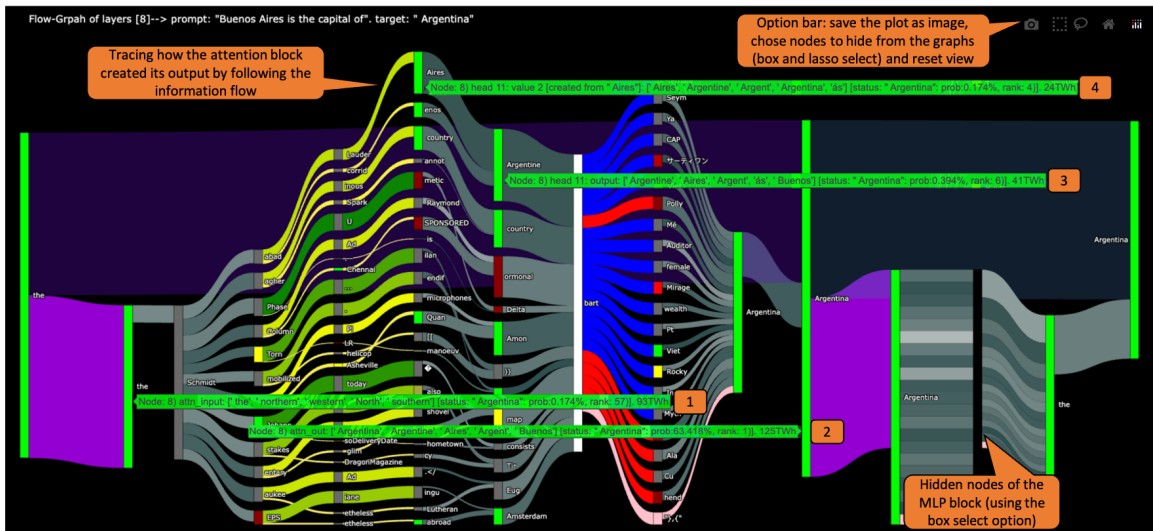

Figure 11: Modeling block number 8 of GPT-2 small for the prompt: "Buenos Aires is the capital of", which the model answers correctly with "Argentina". By using the option bar (top right) we hide the MLP's nodes and focus on the attention sub-block. When hovering over the attention input node (1) a pop-up text window shows information about its corresponding HS, revealing its top projection tokens and how this HS ranks the token "Argentina" (giving it less than a 1% chance). Comparing the input to the output HS of the block (2) we can understand this block promotes the token "Argentina" (the output ranks it with around 63% chance). In order to identify how this block creates its prediction we follow the flow of the model and notice attention head number 11 (3), the one with the largest norm from all the heads (we can see this from the width of its connected edge which is proportional to its norm). Its top projection token is "Argentina" and we want to understand how it was created. To do that, we go along the flow to its memory values (heads are the sum of their memory values). We identify that the memory value that had the largest attention score (4) was created from the input token "Aires" (as shown on the pop-up window). This memory value's 4 most probable projection tokens are "Aires", "Argentine", "Argent" and "Argentina", having high intersection with the most probable tokens of its head's projection and the attention output's projection.

in layer 10). The other information when hovering over each node is its top most probable tokens (a list of tokens) and "status", suggesting its relation with another token, "target", chosen by the user (if given); in particular, its probability and ranking for that token.

- In the attention score calculation we can locate which previous key and value were created by which of the input tokens, since they have the same indexes in the attention memory implementation of GPT-2. We present this information by hovering over the nodes in the attention sub-graph.

- Hovering over an edge presents which nodes it connects to along with information about what it represents, for example: if it is an edge between an attention query and key, it will represent the attention score between them. If the edge represents a summation of one HS into another, the information on the edge will be the norm of the summed HS.

- A user invoking the code can choose the model, the prompt, which layers to present, and a "target" token (recommend to be the actual output of the model for the given prompt).

## B  Walkthrough the Graph Model

The flow-model is an interactive plot. At the top right of the screen there is an option bar that enables to focus on specific parts of the model, by hiding chosen nodes. By examining different blocks and focusing on chosen parts of the graph we gain insights into the predictive mechanisms of the models and how they create their predictions. In Figure 11 we explore how gpt-2 small recalls a factual information, tracing which input tokens created the memory value $v_{ji}$ whose head $A_j$ is responsible for the output of the block (showcasing the patterns we identify in section 3). Similar to subsection 5.1 our findings do not assert that the identified components exclusively control the model's final prediction. Rather, they are recognized as the primary elements responsible for shaping the immediate prediction.

## C  Model Selection

As mentioned, we used GPT-2 medium (355M parameters) as our main case study due to its availability, wide use in previous research, the ability to run it even with limited resources, and the core assumption that characteristics we see with it are also relevant to bigger models. To validate ourselves, we also ran parts of our quantitative analysis with GPT-2 XL (1.5B parameters) with the same setup as we had with the medium model, and observed the same behavior; for example, see Figure 12. For these reasons we believe our analysis and modeling are applicable to general GPT-based models and not only to a specific model.

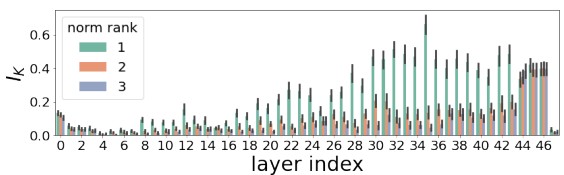

(a) Mean $I_{k=50}$ for only the 3 heads with the largest norm, comparing to attention block output.

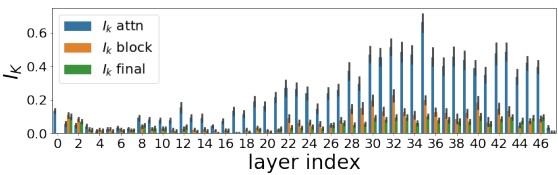

(b) Mean $I_{k=50}$ for only the top-norm head, comparing to attention block output, layer output, and the model's final output.

Figure 12: Projecting attention heads of GPT-2 xl, with the same setups as in Figure 3, shows that the patterns we saw with GPT-2 medium are similar to the ones we see with a bigger model.

## D  Additional Quantitative Analysis of Information Flow Inside the Attention Blocks

### D.1  Additional Setup Information

We provide here additional information on our setup and data selection. The choice of using CounterFact is based on its previous usage in studies on identifying where information is stored in models (Meng et al., 2022, 2023). However, it has the issue that GPT-2 does not succeed in answering most of its prompts correctly (only approximately 8% for GPT-2 medium and 14% for GPT-2 xl), and in many cases, the model's predictions consist primarily of function words (like the token "the").

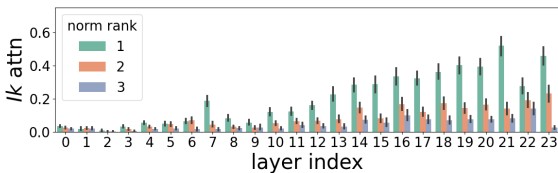

(a) Mean $I_{k=50}$ for only the 3 heads with the largest norm, comparing to attention block output.

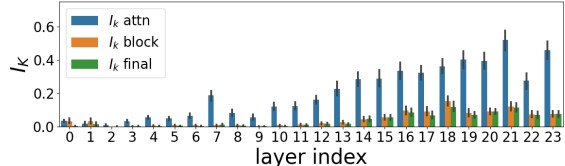

(b) Mean $I_{k=50}$ for only the top-norm head.

Figure 13: Projecting attention heads for prompts from CounterFact, without filtering prompts the model does not answer correctly. These show almost the same results as with the main setup in Figure 3, suggesting the mechanism behind the model's attention works the same for correctly recalling factual knowledge and when predicting tokens of function words.

To avoid editing prompts or analyzing uninteresting cases, we decided to use only prompts that the model answers correctly. A plausible question is whether the model acts differently when it predicts the right answer compared to the general case, without filtering by answer correctness. To examine this we ran our analysis twice, once with only prompts the model knows to answer (like we explain in Section 3) and another time with random prompts from CounterFact. It turns out that the attention mechanism works the same way in both setups, resulting in almost the same graphs (Figure 13), which suggests that the behavior we saw is not restricted to recalling factual knowledge.

The only main difference we notice is the probability score the models give to their final prediction along the forward pass: when the model correctly predicts the CounterFact prompt (meaning it recalls a subject) it starts to assign the prediction high probabilities around its middle layers. However, when the model predicts incorrectly (and mostly predicts a function word), it assigns moderate probabilities starting from the earlier layers (Figure 14). This might suggest for later works to examine if factual knowledge, which is less common than function words in general text, is located in deeper layers as opposed to non-subject tokens.

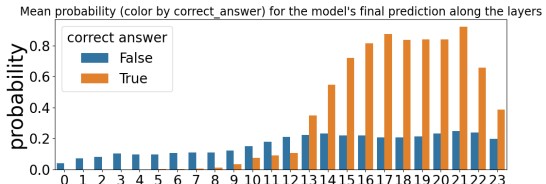

Figure 14: The probability GPT-2 medium assigns to its final predictions' tokens for the projection of the HS between blocks, colored by whether the model returns the true answer or not.

## D.2 Additional Results

We add more graphs to the analysis in Section 3 that help explain our claims in the conclusion of that part. All results are taken from the same experiment we used in that section. Notice that according to the following analysis the model exhibits distinct behavior during its initial 4–6 layers (out of 24) compare to the subsequent layers, as indicated by the low $I_k$ scores for the first layers (Figures 15, 16), a behavior that was noted in previous work (Geva et al., 2022b; Haviv et al., 2023; Dar et al., 2022) and is yet to be fully understood.

Figure 15 illustrates the relationship between the attention output and the residual. It showcases the incremental changes that occur in the residual as a result of the attention updates to it. Similar to how the MLP promotes conceptual understanding within the vocabulary (Geva et al., 2022b), the attention layers accomplish a similar effect from the perspective of the residual. The figure also reveals the high semantic similarity between each attention sub-block and its preceding attention sub-block.

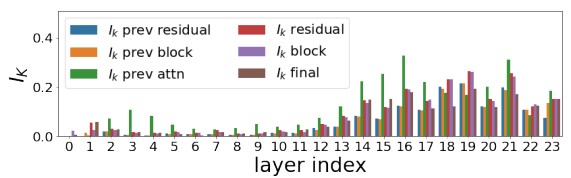

Figure 15: Comparing $I_{k=50}$ of attention output with its current and previous residual (just after it is updated with the attention output) and the block output (note that the input of the attention sub-block is its previous block output). The intersection between the attention output is considered high, which means that the attention sub-blocks have overlapping semantics between different layers.

Figure 17 demonstrates that the information flow we saw from the memory values to the heads output is a behavior that applies to all heads.

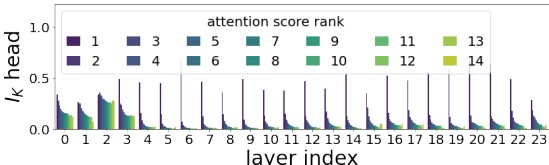

Figure 16: Comparing $I_{k=50}$ of memory values with the output of their heads, according to the memory value norm rank compared to other values in the same head (the complete analysis behind Figure 5). This example claims that the semantics of each head is determined by its top memory value since only the top 1–3 memory values have some semantic intersection with their heads (starting from the 4-th layer) and the rest of the heads have almost no intersection (the number 14 suggest that the longest input we used for this experiment was 14 tokes).

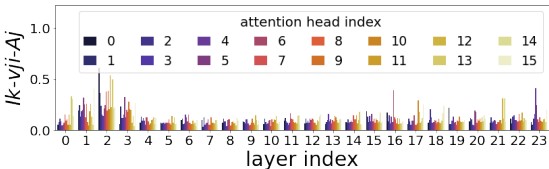

Figure 17: Comparing $I_{k=50}$ of memory values with the output of their heads, according to head indices. This shows that there are no particular heads that are more dominant than others (after the first few layers).

Figure 18 demonstrates the alignment in projection correlation between each input token and its corresponding memory values. For every memory value $v_{ji}$, we examine the probability of its input token (the $i$-th input token) after applying a logit lens to $v_{ji}$. Our underlying assumption is that if the generated values share common semantics, then the probability of the input token should be higher than random (which is nearly 0). The results substantiate this assumption, revealing higher scores in the subsequent layers.

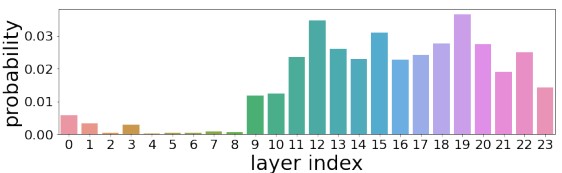

Figure 18: The probability of input token in the vectors of memory values they generated.

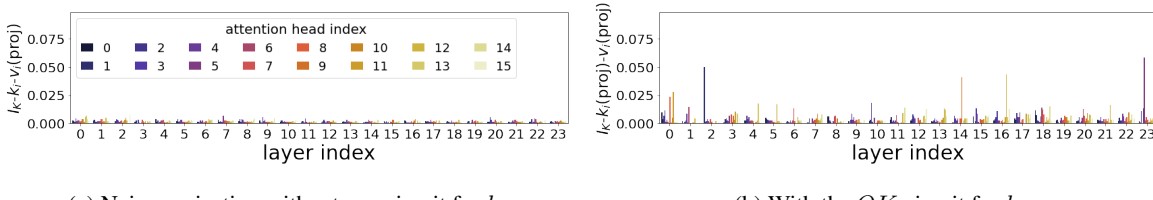

(a) Naive projection without any circuit for $k_i$.

(b) With the $QK$ circuit for $k_i$

Figure 19: Comparing $I_{k=50}$ token projection alignment between the head outputs of $W_K$ and $W_V$ ($k_i$ and $v_i$), with and without the $QK$ circuit for $W_K$ ($v_i$ is projected with $W_O$ and can be see as the output of the $OV$ circuit).

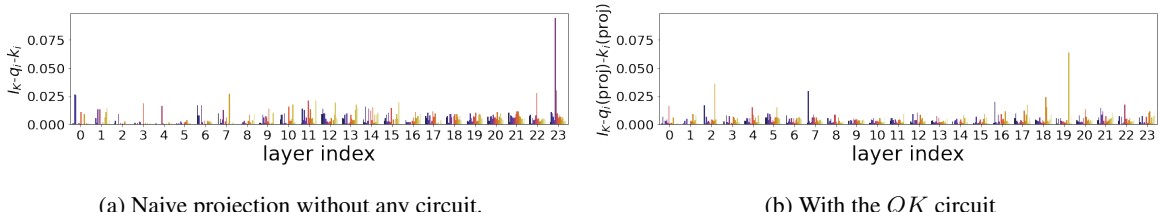

(a) Naive projection without any circuit.

(b) With the $QK$ circuit

Figure 20: Comparing $I_{k=50}$ token projection alignment between the head outputs of $W_Q$ and $W_K$ ($q_i$ and $k_i$), with and without the $QK$ circuit. The mean intersection for two random sampling of 50 items (without duplication) from a set the size of GPT-2 vocabulary (50257) is around 0.05 matches, which equals to $I_{k=50}$ of 0.001. However, when we apply the logit lens to two random vectors, it is observed that due to certain biases in the decoding process, the average $I_{k=50}$ value is 0.002 .

### D.3 Are All HS Interpretable? Examining the $QK$ Circuit

Similar to our analysis of the attention matrices $W_V, W_O$ (section 3), we try to find alignment between $W_Q, W_K$ outputs and other HS of the model. The work of Dar et al. (2022), who first projected the matrices $W_Q, W_K$, emphasizes the importance of projecting the interaction between the two using the $QK$ circuit, meaning by projecting the matrix $W_{QK} = W_Q \cdot W_K$. Using the data from section 3, we collected dynamic HS that these matrices generate, $q_i$ and $k_i$ (attention queries and keys), to examine their alignment between each other and between the memory value $v_i$ they promote (each $k_i$ leads to a single $v_i$, noting we already saw the latter is aligned with the attention's and model's outputs section 3). We project $q_i$ and $k_i$ using two methods: once with the naive logit lens ($LL$) and once using the $QK$ circuit, by first multiplying $q_i$ with $W_K$ ($LL(q_i \cdot W_k)$) and $k_i$ with $W_Q$ ($LL(W_Q \cdot k_i)$). Our hypothesis was that we will see some overlap between the top tokens of $q_i, k_i$ and $v_i$; however, the results in Figures 19, 20 show almost no correlations using both methods, in contrast to the results we saw with $W_V$ and $W_O$ (subsection 3.1).

We believe there are two options for the low scores we see. The first option is that $W_Q$ and $W_K$ deliberately promote different tokens, with no alignment between $W_Q, W_K, W_V$. The idea behind that is to check the associations between different ideas (for example, an unclear association can be a head's keys $k_i$ with meanings about sports but with values $v_i$ about the weather). Another option is that the output of $W_Q, W_K$ operates in a different embedding space, which is different than the rest of the model, explaining why logit lens would not work on it. A support for this idea can be the fact that the output of these matrices is not directly summed up with the residual, but is only used for computing of the attention scores (that are used as coefficients for $v_i$, which *are* summed into the residual).

In our flow graph model, the user can chose to merge $q_i, k_i$ nodes into one with $v_i$, making them less visible. However, we decided to display them by default and to project them with the $QK$ circuit, since during our short qualitative examination we noticed examples that suggest that the first option we introduced might be true. In Figure 6b we can see that the projection of the key with the highest attention score behind the Negative Name Mover Head holds the meaning of "Mary". In this case, we can imagine that the model implements a kind of if statement, saying that if the input has really strong semantics of "Mary", we should reduce a portion of it (maybe, to avoid high penalty when calculating the loss during training).

# E Layer Norm Uses as Sub-Block Filters

We present additional results about the role of LN in changing the probabilities of each sub-blocks' input, including results for both LN layers in GPT-2. Tables 1 and 2 show the top tokens before and after $ln_1$ for two different layers. Figure 21 gives a broader look at the effect of $ln_1$, detailing some examples across layers in Table 3. We repeat these analysis with $ln_2$ in Figure 22 and Tables 4 and 5.

We include an example about the LN effect on the HS if the projection was done without the model's final LN, $ln_f$, which is attached to the decoding matrix. Initially done to examine the effect of $ln_1$ and $ln_2$ without $ln_f$ on projection, the results in Figure 23 and Table 6 highlight the importance of using $ln_f$ as part of the logit lens projection, since the tokens we receive otherwise look out of the context of the text and tokens our model promotes in its generation.

| $ln_1$ **5** | $ln_1$ **11** | $ln_1$ **17** | $ln_1$ **23** |
|---|---|---|---|
| the | the | the | the |
| using | not | North | a |
| not | a | Google | English |
| this | T | a | India |
| within | C | South | Russian |
| in | U | company | German |
| , | in | now | North |
| and | , | Germany | South |
| : | which | not | " |
| outside | N | still | K |

Table 3: Tokens that lose the most probability after $ln_1$, as collected from the experiment in section 3. Earlier layers' LNs demote more tokens representing prepositions than later layers.

| **before** $ln_1$ | **after** $ln_1$ |
|---|---|
| English | English |
| the | Microsoft |
| Microsoft | abroad |
| North | subsidiaries |
| not | North |
| abroad | combining |
| a | downtown |
| London | Redmond |
| India | origin |
| origin | London |

Table 1: The top tokens before and after $ln_1$ at layer 15, according to the mean HS collected in section 3. We can see how the LN filters all the function words from the 10 most probable tokens while introducing instead new tokens like "Redmond" and "downtown".

| **before** $ln_2$ | **after** $ln_2$ |
|---|---|
| the | English |
| not | Microsoft |
| English | not |
| abroad | abroad |
| Microsoft | subsidiaries |
| a | origin |
| origin | combining |
| T | the |
| U | photographer |
| photographer | renowned |

Table 4: The top tokens before and after $ln_2$ at layer 13.

| **before** $ln_1$ | **after** $ln_1$ |
|---|---|
| the | abroad |
| not | Microsoft |
| abroad | subsidiaries |
| a | combining |
| origin | English |
| Microsoft | origin |
| T | not |
| Europe | Europe |
| U | photographer |
| C | the |

Table 2: The top tokens before and after $ln_1$ at layer 13.

| $ln_2$ **5** | $ln_2$ **11** | $ln_2$ **17** | $ln_2$ **23** |
|---|---|---|---|
| the | the | the | the |
| in | a | Google | a |
| a | T | French | German |
| using | U | Boeing | North |
| , | in | company | South |
| and | C | a | K |
| : | , | London | " |
| now | : | not | N |
| this | and | sports | Kaw |
| at | at | hockey | Boeing |

Table 5: Tokens that lose the most probability after $ln_2$, similarly to Table 3.

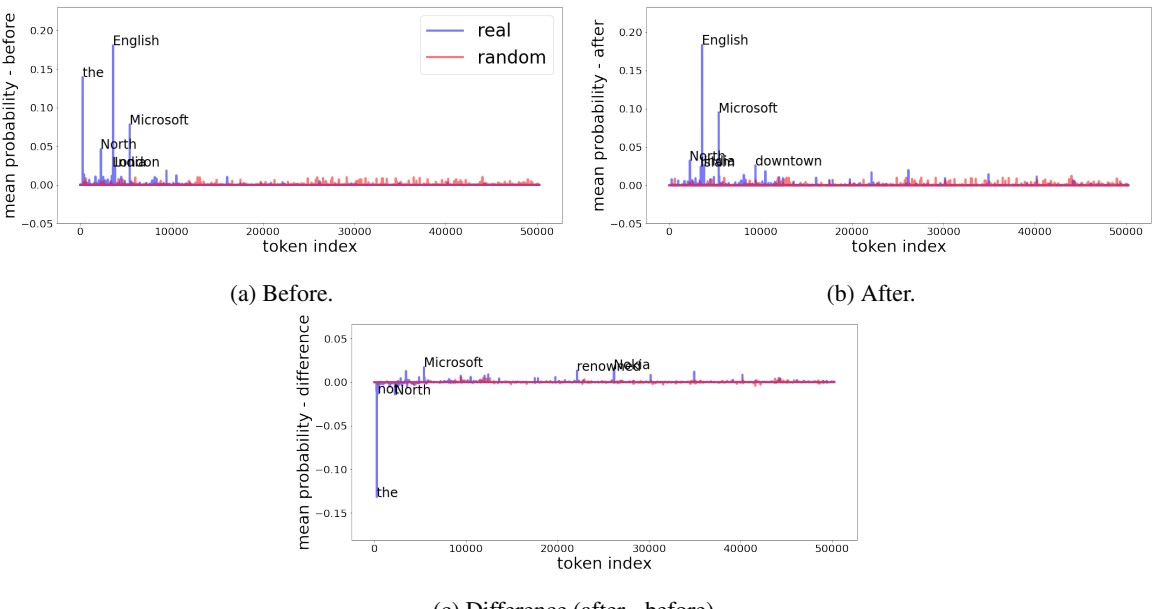

(a) Before.

(b) After.

(c) Difference (after - before).

Figure 21: The probability of all the tokens in GPT-2 before and after the first LN, $ln_1$, in layer 16, including annotation for the largest-magnitude tokens. We see the difference in the distributions of tokens between randomly generated vectors and the ones we sample from CounterFact, which we find reasonable when answering factual questions. Especially if the questions are about a finite number of domains, the network promotes tokens not in a uniform way (like the random vectors does). Although tokens like "Microsoft" and "The" have a high probability before the LN, while the first gained more probability during the process the second actually loses, suggesting this is not a naive reduction to the probable tokens at each HS.

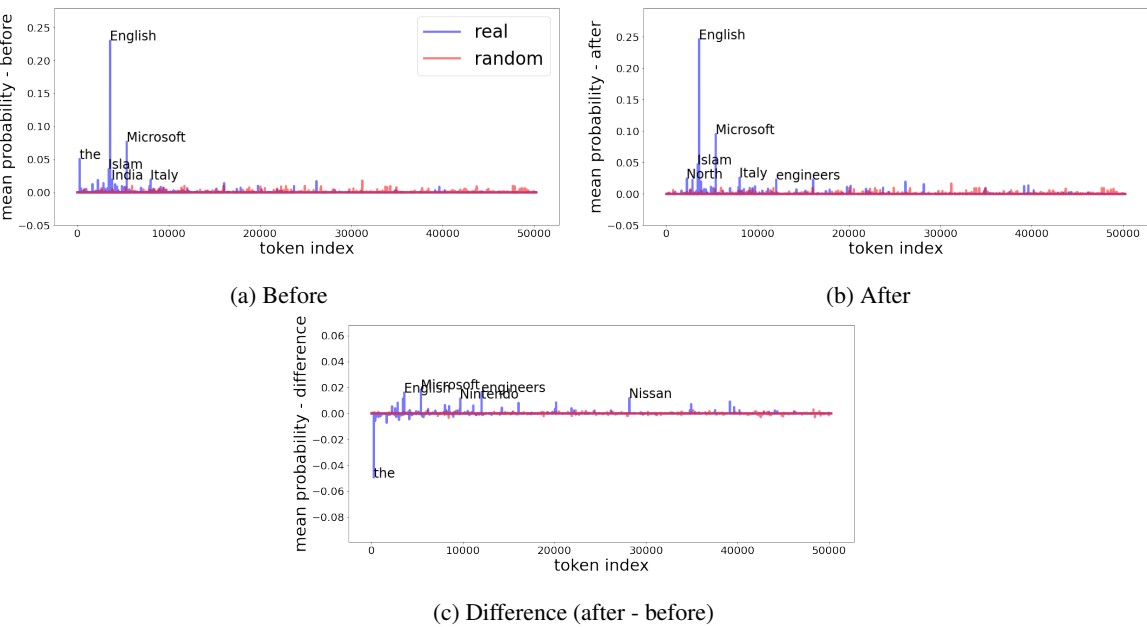

(a) Before

(b) After

(c) Difference (after - before)

Figure 22: The effect of $ln_2$ at layer 16 on tokens' probabilities. We can see similar highly probable tokens as in Figure 21, since the only difference between the inputs of $ln_1$ and $ln_2$, which is the residual stream, is the attention output of that layer (which is known to be gradually and does not steer the probability distribution dramatically Figure 14, 15).

| $ln_1$ **5** | $ln_1$ **11** | $ln_1$ **17** | $ln_1$ **23** |
|---|---|---|---|
| Zen | not | the | the |
| imperialist | the | English | , |
| Sponsor | Europe | a | " |
| abroad | C | " | - |
| mum | abroad | football | ゼウス |
| utilizing | T | and | externalToEVAOnly |
| UNCLASSIFIED | pure | Toronto | sqor |
| conjunction | English | sports | quickShipAvailable |
| tied | ized | first | 龍契士 |
| nineteen | Washington | - | ÃÂÃÂÃÂÃÂ |

Table 6: Top tokens that lost probability after applying $ln_1$ when projection is done without $ln_f$.

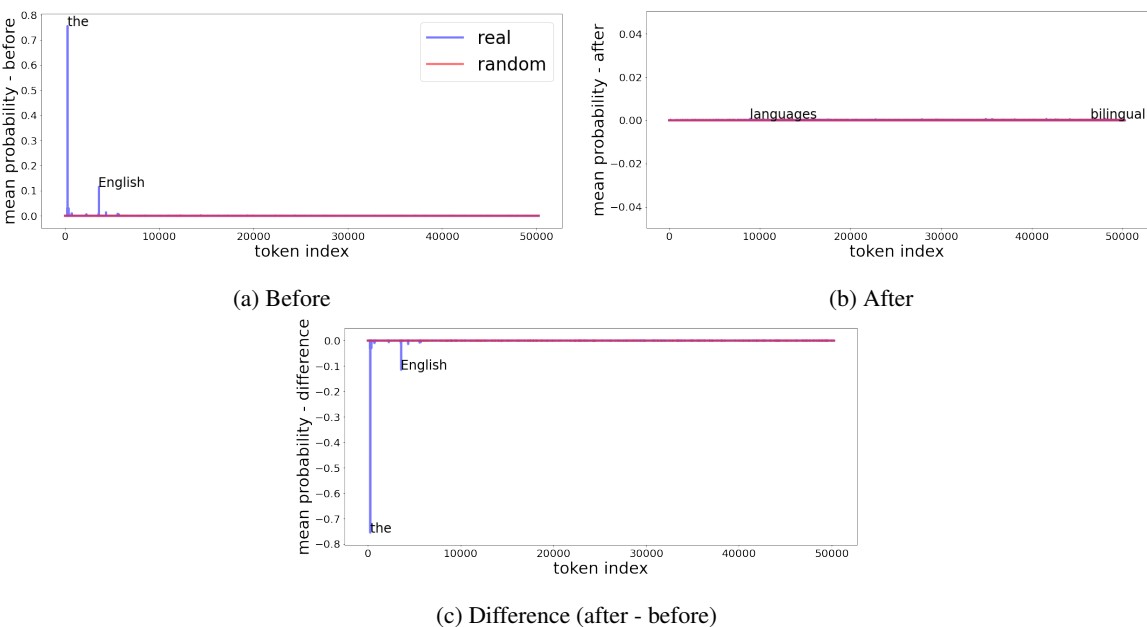

(a) Before

(b) After

(c) Difference (after - before)

Figure 23: The affect of $ln_1$ at layer 16 on tokens' probabilities (similar to Figure 21), but when the projection is done without $ln_f$.