# OpenReview forum: "VISIT: Visualizing and Interpreting the Semantic Information Flow of Transformers"
_EMNLP/2023/Conference — EMNLP 2023 Findings_

### Official Review · Reviewer_arzY · 2023-07-26

**Typos Grammar Style And Presentation Improvements:** 1. It seems that there is quite a bit…
**Soundness:** 3

**Excitement:**

4: Strong: This paper deepens the understanding of some phenomenon or lowers the barriers to an existing research direction.

**Paper Topic And Main Contributions:**

This paper investigates LM attention heads, memory values, and the vectors dynamically created and recalled by the models while processing a given input. The authors further develop a technique to visualize the forward pass of GPTs as an interactive flow graph. This proposed visualization tool reveals new insights about the role of layer norms as semantic filters that influence the model output, and about neurons that are consistently activated during forward passes, acting as regularization vectors.

**Questions For The Authors:**

1. Why was GPT-2 chosen as the backbone model for visualization? Does GPT-2 have unique and extensive features that make it the best choice for building a visualization tool? Although the power of the GPT series is well-known, it is hoped that the authors can systematically elaborate on the motivations for this work.
2. How to generalize this work to other types of language models?
3. Could the author provide more quantitative validation regarding the findings mentioned in the text?
4. This paper uses the lens of logits, which transform intermediate embeddings into a probability distribution over the model's vocabulary using a decoding matrix. However, the employment of the top-k operation can lead to information loss and incorrect interpretation. Furthermore, tracing the disappearance and appearance of words during the model's inference stage proves challenging, as the representative words for certain intermediate embeddings appear unrelated to the words in the subsequent layer. More discussion on this issue is recommended.

**Reasons To Accept:**

1. This work might significantly impact the explainability research of NLP and DL communities.
2. The theoretical analysis part is thorough. It appears that the authors have conducted sufficient research on GPT-related works.

**Reasons To Reject:**

1. The experimental part needs to be more sufficient. The authors need to devise more persuasive comparative experiments to verify the superiority of the proposed method.
2. The author needs to emphasize academic innovation, clearly explaining their work's principles. An excellent academic paper is always inspirational; it can motivate people to explore more.
3. It is suggested that the author redesign the structure of the paper. The current structure of the article is more like a technical report than an academic paper.

**Reproducibility:**

4: Could mostly reproduce the results, but there may be some variation because of sample variance or minor variations in their interpretation of the protocol or method.

**Reviewer Confidence:**

3: Pretty sure, but there's a chance I missed something. Although I have a good feel for this area in general, I did not carefully check the paper's details, e.g., the math, experimental design, or novelty.

---

> ### Author Rebuttal · Authors · 2023-08-28
>
> First of all, thank you for the review. We are glad that you see a potential in our work to significantly impact the explainability research.
>
> >"Why was GPT-2 chosen as the backbone model for visualization?" and "How to generalize this work to other types of language models?"
>
> GPT-2 is only used as our test-case as an example for decoder-only LMs (similar to the main works in this field of intepretability of language models, such as  Geva et al. (2021, 2022a, 2022b), Dar et al. (2022)), but the visualization is not limited to it, and in fact with the release of our code we also support GPT-J and provide guidance on how to adapt the tool to any decoder-only LM.
> With that being said, our purpose was to present a new method and it can be further adapted to other types of Transformers. We simply worked with the most common architecture and models in works similar to ours.
>
> >"Could the author provide more quantitative validation regarding the findings mentioned in the text?" and "The experimental part needs to be more sufficient"
>
> We can conduct more experiments but we are not sure to which of our current results you referred (especially because of the amount of space that the already conducted ones take in this paper).
> Please note: the results from **Section 3** are a continued discussion of the works of Elhage et al. (2021),  Geva et al. (2022b) and Dar et al. (2022), and we just tried to fill the gap concerning the importance of specific components of the attention sub-blocks to better design our visualization (also note we conduct additional experiments regarding **Section 3** which are included in the **Appendix**).
> The results from **Section 5**, although we continue to investigate some of them in the appendix, are shown briefly to illustrate the power of our tool.
> This also explains why we chose to structure our paper in the way that we did: we first establish the information flow with quantitative results in **Section 3**. This, with knowledge from prior work, guides our flow graph construction method in **Section 4**, which facilitates the analyses in **Section 5**.
>
> >"This paper uses the lens of logits... the employment of the top-k operation can lead to information loss and incorrect interpretation"
>
> We totally agree with this statement and we even mention a warning about it in the Limitations section regarding the $I_k$ metrics.
> The top-k usage is a core part of LogitLens (usually with very small k, i.e., top-1) and we in fact engage with the issue more than in most prior papers.
> As we share this concern, we will incorporate it into the paper.
>
>
> >"The author needs to emphasize academic innovation... motivate people to explore more"
>
> As we try to present our work in a modest way, we believe our flow-graph modeling (the tool) can be a booster for many research questions/directions, just like you recognized and mentioned. Our claim is that qualitative examinations of those models using our new tool, which is highly different than previous methods, can make a huge impact (and we try to support this by presenting our findings in **Section 5**).
> Given your feedback, we will highlight this point earlier at the beginning of the paper.
> We believe that with more opportunities to see results of our analysis approach, the inspiration and motivation will be more clear (according to our experience of presenting the tool to colleagues). We have produced more valuable plots but unfortunately due the conference rules we can't add more plot examples at the moment.
>
>
> We want to take this chance to add a more tangible example of how we believe this tool can help research (and why we think you might also be excited by it). For that, we will use **Figure 11** from the paper: when we investigated its prompt, we first looked at the full graph that included multiple blocks. By focusing on different components we found the specific block where the model first predicts what becomes its final prediction (in the Figure we present only this block). Focusing on this block allows us to see which heads' neurons are the ones that bring this prediction as detailed in its caption (including more discoveries), without any previous knowledge about those neurons (on-the-fly). This is just one example of a new level of interpretability that can boost any research in the field with very little technical or previous knowledge (i can imagine how many students will find this useful).
>
> Thanks for all the great questions and notes. Please let us know if you have any remaining concerns that we can answer and perhaps change your evaluation.

---

### Official Review · Reviewer_mg6v · 2023-08-03

**Soundness:** 4

**Excitement:**

4: Strong: This paper deepens the understanding of some phenomenon or lowers the barriers to an existing research direction.

**Paper Topic And Main Contributions:**

The paper introduces a visualisation tool that creates a flow graph of different hidden states within GPT2 by projecting them to the Language Model head, a process called logit lenses. The paper goes one step further than previous work by not just projecting the hidden state at the end of each attention block but rather doing it even at the head level by looking at head-specific dimensions when projecting to the LM head, albeit this part was a bit unclear. This obtains very fine-grained interactions within each attention block represented by tokens instead of dense hidden states.

By utilising this method the authors claim three main contributions:
(1) reflect the mechanistic analysis of Wang et al. (2022) on indirect object identification in a simple way; (2) analyze the role of layer norm layers, finding they act as semantic filters; and (3) discover neurons that are always activated, related to but distinct from rogue dimensions (Timkey and van Schijndel, 2021), which we term regularization neurons.

**Questions For The Authors:**

Does the code release include the full visualisation tool? Ie. a library where I can query my model (I guess restricted to GPT2) and obtain a flow graph as in Figures 1, 5 and 6, and the functionality the appendix showcases?

What should one learn from section 3? While 3.4 tries to explain it, I am still unsure whether anything new is learned from it. Maybe that some heads carry most of the "semantic meaning". Still two pages seem too much.

Could you perhaps justify why this paper should be presented as a "Interpretability, Interactivity, and Analysis of Models for NLP" conference paper and not as a demo of the tool at EMNLP?

**Reasons To Accept:**

1. The main reason to accept would be the release of the tool to create the flow graphs, which may aid future research. This is especially true for the most useful parts of the flow graph, which would be the less fine-grained ones, as will be discussed later.
2. Of the three main contributions, the one that strikes as new and interesting is the observed phenomena of semantic filters by layer norm layers. It is interesting how function words contribution is decreased after applying them, highlighting the utility of these layers to increase the weight of content "words" (or at least hidden states close to those words.). I would have enjoyed more quantitative results regarding this.

**Reasons To Reject:**

1. The method proposed seems too fine-grained. It works when one already knows what to look for. Each flow graph is focused on a single layer. The projection of specific heads (either queries, keys or values) to the subdimensions of embeddings at the LM head (ie. j:j+h/d as in line 236) doesn't seem to be that useful. The fact it needs to be projected through W_O and that no quantitative results to properly measure  any patterns or useful information from these projections, hinders the overall approach. Moreover, the projections for each head are not properly explained, one has to deduce just from line 236 that each hidden state is projected through W_O and then to the corresponding dimensions of the LM head, not just for A_j but also for keys, queries and values. While "full" hidden states or weights that share the full dimension (d) are well projected to specific tokens as shown by work such as Dar et al. (2022), the new contribution of this work which is looking at the head level is only demonstrated through a couple of examples and histogram figures, but I am not sure what one really learns other than that certain heads have higher norms and therefore their influence in the output of the attention block is bigger. What is the contribution of the visual flow there? When I look at Figure 1 and 5, I do not see any meaningful "projections" at the head level. Its utility is only portrayed when one already knows of a specific head phenomena from other work, as shown in Figure 6. And while some plots help grasp some degree of interpretability outside the head level (ie. layer norm, attention block, feed-forward, residual, ie, hidden states that share the same dimension), this was already shown by previous work. Still, a tool to properly visualise these flows, with top tokens for each hidden state is very nice, and considered a reason to accept.

2. The results themselves are unsurprising. While this by itself is not a reason to reject, combined with the fact that other works have already discussed most of the points the paper sort of hinders the "reasons to accept". This can be seen as a reason not to accept, rather than to reject.

3. Why only GPT2? How can we know this tool will be useful in future research if it was only tested on a single model? If the restriction is that only decoder systems benefit from this visualisation, why not try other decoder only models and show the flexibility of this approach? It seems that most of what is discussed in the paper was already known for GPT2, so if the point is to present a new tool that helps in explainability of transformer models, as the title would suggest, showcasing other models would be important.

Overall I feel like the paper is trying too many things at once. It presents as a tool for visualising the flow of information, and gives some nice examples on where it is useful, but at the same time focuses on contributions which were already known or a bit redundant/expected, with qualitative experiments/examples. In truth, the tool itself should be the main contribution, and less words could be devoted to try to justify new contributions in a shallow way, which are not that surprising anyways, while more relevant examples on how the tool works are relegated to the Appendix. To sum up, if the paper is trying to show how information flows in the model, other papers have already done so in a more clear and concise way, however, doing so through a visual tool projecting to specific tokens to help interpretability is novel.

**Reproducibility:**

2: Would be hard pressed to reproduce the results. The contribution depends on data that are simply not available outside the author's institution or consortium; not enough details are provided.

**Reviewer Confidence:**

4: Quite sure. I tried to check the important points carefully. It's unlikely, though conceivable, that I missed something that should affect my ratings.

**Typos Grammar Style And Presentation Improvements:**

Check grammatical errors in the figures. Attention seems to be spelled wrong (attentnion) in all of them.
There needs to be more clear formulas. At the current stage, the logit lens explanation is confusing, as it is applied differently at head level and before/after W_O projection. The formulation is not clear when you have a head with dimension d/h. The W_O projection seems to imply that the same is done when projecting to D, ie. using only j to j+d/h, but it should be clarified. Moreover, are query and values as shown in the figures also projected to W_O? Overall I think all the formal definitions can be somewhat improved to help the reader.

---

> ### Author Rebuttal · Authors · 2023-08-28
>
> Thank you for your review. We are glad you see the merit of our flow-graph modeling contributing to future research.
>
> We want to start with the first point from the reasons to reject since it includes many concerns that we believe that after they are cleared, your perception of our work might change:
>
> >``The method proposed seems too fine-grained... When I look at Figure 1 and 5, I do not see any meaningful "projections"... Its utility is only portrayed when one already knows of a specific head phenomena''
>
> This is exactly what we aim at solving with our new method, and we establish its correctness with the experiments in **Section 3** and **5.1**!
> Let's take for example **Figure 5**: when we investigated its prompt, we first looked at the full graph that included many blocks (i.e., multiple blocks in one plot). By focusing on different components (using the select options from the top right bar) we found the specific block where the model first predicts what becomes its final prediction (in the figure we present only this block). Focusing on this block allows us to see which heads' neurons are the ones that carry this prediction as detailed in its caption (including more discoveries), while all that is being done without any previous knowledge about those neurons (on-the-fly). This example is not hand-picked since it applies to many more cases, such as **Figures 6** and **11**. Let us put some excitement to start this rebuttal: this is just one example of a new level of interpretability that can boost many research questions/directions in the field (we are not aware of any similar approach regarding the ability to reveal such amount of information).
>
> >``What should one learn from section 3'' and the concern about the projection from the first reasons to reject
>
> We are not the first to project attention components through circuits ($W_O$). In fact, **Section 3** is a continued discussion of Dar et al. (2022) and Elhage et al. (2021) who were the first to so. Both explained why and how to project these components using similar methods and we mainly continued and conducted experiments to prove it for our cases in **Section 3.1** (those are indeed quantitative ones, and please also refer to the Appendix where we further discuss that: **Additional Results** and **Are All HS Interpretable? Examining the $QK$ Circuit**).
> We realize that it might be hard to understand this without deeply knowing those previous works and we would add an additional technical explanation about this.
>
> So, what is the difference between us and previous works (or more correctly, why is **Section 3** so long and what we can learn from it): our main contribution here is that we did this in scale at dynamic hidden states (rather than static as in Dar et al.), identifying with interpretability methods where each ``meaning'' comes from (**Sections 3.2, 3.3**). This is a part we found to be missing in the literature and it has an important purpose in supporting the rest of the paper, since it explains why we can filter some of the attention components in our visualization. You mentioned these parts were not surprising and we would argue this is actually a strength(!). Of course it's known that some heads may be more important than others, but identifying them on-the-fly according to their norms like we did is an innovation.
>
> >About the code release (your first question and also about Reproducibility)
>
> Our tool is Python based and we provide implementation details in the appendix (**Implementation Details and how to Read the Graph**). Of course, we will release the full tool.
> Currently we provide the setup for GPT-2 and GPT-J, and also a guide (notebook with instructions) on how to adjust it to other models with only moderate changes.
> The tool wraps those models so for every input you can inference, it is straightfoward to display its corresponding flow-graph (and to do so in a few seconds even on simple setup like Colab without any GPU). With a simple configuration you can select which blocks to present on a single plot (you can display multiple blocks on a single graph. In the paper we present only single ones due to lack of space).
> We do regret in not sharing more examples with this submission since we think it could make our methods more tangible, but unfortunately we can't do this now due to conference rebuttal rules.
>
> >``Could you perhaps justify why this paper should be presented as a "Interpretability, Interactivity, and Analysis of Models for NLP"''
>
> Considering our analysis (**Section 3**) and the research findings we produced from the tool (**Section 5**), we believe we can contribute more by presenting our work under the interpretability track, rather than the demo track, which less appropriate to presenting our analysis.
>
> >``Why only GPT2?''
>
> As we mentioned, the tool is not limited to GPT-2.
> We indeed used GPT-2 as our test-case (mainly for **Section 3**, which we further explain in **Model Selection**), as we follow previous works in this field that also used only GPT-2 as a decoder transformer to analyze (Geva et al. (2021, 2022a), Dar et al. 2022), or  created a toy example decoder-only LM and used it as their only test-case (Elhage et al. 2021). This line of work demonstrated GPT-2 is a good proxy for auto-regressive interpretability research. But again, we've also provided the setup for GPT-J and will release example plots in the final version.
>
> >Finally, we want to address your concerns about the structure of the paper:
>
> The structure we chose for the paper is to (1) fill in the missing knowledge we found necessary for the presentation of our tool (**Section 3**), (2) then presenting the tool (**Section 4**, notice how short is the technical part in the main paper) and (3) demonstrating how it can be used (**Section 5**, including new findings you mention as interesting). Only presenting the tool would not be ideal since it would lack the ability to support its correctness in reflecting the inner computation of LM, and will miss some immediate findings the community for sure can benefit from. Wrapping everything in 8 pages was not easy but we tried our best.
>
> Again, we want to thank you for your review and hope we answered all your concerns so that you may consider changing some of your scores. We believe that with more opportunities to work with our tool we could reflect its power in a more convincing way. Please let us know if we can add more clarifications about our work.

---

### Official Review · Reviewer_PqVU · 2023-08-11

**Soundness:** 3

**Excitement:**

3: Ambivalent: It has merits (e.g., it reports state-of-the-art results, the idea is nice), but there are key weaknesses (e.g., it describes incremental work), and it can significantly benefit from another round of revision. However, I won't object to accepting it if my co-reviewers champion it.

**Paper Topic And Main Contributions:**

* *Contribution 1:* This work studies the information flow in a GPT model by looking at the attention module using an existing method, LogitLens.

* *Contribution 2:* With the analysis results, the authors propose a tool to visualize the information flow and conduct three case studies to verify the usability of the tool.

**Questions For The Authors:**

* Since the code is not included, how is the applicability of the tool to other models and datasets?
* How is your visualization tool related to other lines of work that model the information flow in attention? (https://aclanthology.org/2020.acl-main.385.pdf, https://aclanthology.org/2021.acl-short.8.pdf)

EDIT: Both questions are addressed by the authors in the response.

**Reasons To Accept:**

* This work proposes a tool to visualize the simplified information flow. Tools like this (if accessible) can facilitate research in understanding a model's decision.
* Although the analysis part is only based on a single dataset (CounterFact) and a single model variant (GPT), one of the case studies shows the applicability by checking the finding through their visualization tool aligns with the finding in a previous work.

**Reasons To Reject:**

* The core method used for analysis is *LogitLens*, which was originally proposed in a well-known blog post, but it was shown to be brittle in a  recent work (https://arxiv.org/pdf/2303.08112.pdf), as also cited by the authors. The authors claim that the basic approach of LogitLens is applied because they are interested in the interim hypothesis instead of the final layer’s output, which may not be convincing enough because the Din et al. paper also presented differences between the original LogitLens versus the improved version.
    * EDIT: As mentioned in the authors' response, I think this is less of a concern because (1) The paper I mentioned was published too closed to the submission deadline, and (2) there are components where applying these methods would be difficult/inefficient.
* As mentioned in the strength section, with results shown from a single dataset and a single model variant can be a weakness, although not necessarily a reason to reject.
* The issues of the proposed visualization tool. (These are more of presentation issues, but putting in this section because the visualization tool is the core contribution of this work)
    * EDIT: This should've been addressed per the author's response.
    * More careful consideration such as colorblind friendliness or contrast level should be taken towards the interface design of the visualization tool.
    * As also mentioned by the authors in the Limitation section, the approach prunes out neurons that are less activated for better visualization, but it has to be under the hypothesis that the more activated neurons are more important for prediction, which may lead to misleading results if this hypothesis fails to hold.

**Reproducibility:**

4: Could mostly reproduce the results, but there may be some variation because of sample variance or minor variations in their interpretation of the protocol or method.

**Reviewer Confidence:**

3: Pretty sure, but there's a chance I missed something. Although I have a good feel for this area in general, I did not carefully check the paper's details, e.g., the math, experimental design, or novelty.

**Typos Grammar Style And Presentation Improvements:**

* Figure 2 a & b: Consider decreasing the maximum $I_k$ values to be shown. Currently, all the bars in 2(a) look squashed into x-axis. It is difficult to see the claim that no correlation between $I_k$ and layer index before $W_o$ projection can be found.
* Figure 7: some words in the figure are overlapped with each other.
* There are some tense inconsistencies in different sections (e.g. Sec 3.3, 5.3).

---

> ### Author Rebuttal · Authors · 2023-08-28
>
> We want to thank you for your detailed review and we hope to highlight some of our work strengths compared to similar works.
>
> >how is the applicability of the tool to other models and datasets
>
> As we mentioned in the implementation details **Implementation Details and how to Read the Graph** the tool is Python based and can be adapted to any decoder-only LM. We provide the setup for GPT-2 and GPT-J, and also a guide (notebook with instructions) on how to adjust our code to other models with only moderate changes.
> The tool is not limited to any dataset or text, just like we used it in **Figure 6**. If a model can inference a prompt - the tool can display it.
>
> We want to take this chance to add an example of how we believe this tool can help with different models, datasets, and research questions, like identifying sub-components that store or control a behavior. For that we will use **Figure 11** from the paper: when we investigated its prompt, we first looked at the full graph that included multiple blocks. By focusing on different components we found the specific block where the model first predicts what becomes its final prediction (in the paper we present only this block). Focusing on this block allows us to see which heads' neurons are the ones that carry this prediction as detailed in its caption (including more discoveries), without any previous knowledge about those neurons (on-the-fly). Let us put some excitement to start this rebuttal: this is just one example of a new level of interpretability that can boost many research directions in the field.
>
> >How is your visualization tool related to other lines of work that models the information flow in attention?
>
> Thank you for the citations on attention flow. These studies propose algorithms for quantifying the contribution of input features (words) by aggregating attention scores. They thus have a very different goal. In contrast, we interpret intermediate processing steps in the model, and we are doing this with different methods. We will add this distinction to the related work section in our revision.
> In related to other works we cited in **Introduction** and **Related Work**, we claim that our visualizations are more intuitive and, in an innovative way that does not require any prior knowledge, we add interpretability into the visualizations.
>
> >The core method used for analysis is *LogitLens*... but it was shown to be brittle in a recent work
>
> We understand your concern, and indeed LogitLens is not perfect, but please note the following:
> The works you mentioned, Belrose et al. and Din et al., investigate the hidden states between blocks (that is, activations, as a dynamic analysis). By learning a linear/affine transformation, on large amounts of example activations, they find ``shortcuts'' in the projection of the logits. Those works mostly support the well known claims that GPT-like models have redundant calculations (which translates into early exit methods, just like they showed).  However, we analyze various components in the model that were not considered in those two studies: both dynamic components like outputs of attention heads/vectors and static components like neurons in $W_O$ and $FF_1$, $FF_2$. The learnable projections cannot be easily applied to such components (if at all possible) so it hinders the options of projecting every neuron in an efficient way instead of relying on more known and basic method like LogitLens (both Belrose et al. and Din et al. were published just 3 months before the submission deadline).
>
> >The issues of the proposed visualization tool... More careful consideration such as colorblind friendliness
>
> The tool's colors are highly adjustable. We will be more than glad to share a colorblind-friendly configuration when we publish it.
>
> >approach prunes out neurons that are less activated for better visualization
>
> Indeed, our pruning approach works under the hypothesis that highly activated neurons are more important for the prediction. We will clarify this in the revision. We believe this is a reasonable hypothesis, for the following reasons: First, we first establish the part about investigating the flow of information in the attention sub-blocks (Section 3). Second, we refer to several studies that made similar discoveries for the MLP sub-blocks. Third, we also show neurons that are least activated as a control. We are aware the assumption might not fully hold in every case. However, as a general approach and given the inability to present too many nodes in a single graph, we find that our pruning approach is effective for reflecting most of the interesting components of the model.
>
> >with results shown from a single dataset and a single model variant can be a weakness
>
> We indeed used a single dataset with GPT-2 as our test-case (mainly for Section 3, which we further explain in **Model Selection** and **Additional Setup Information**), as we follow previous works in this field that also used only GPT-2 as a decoder transformer to analyze (Geva et al. (2021, 2022a), Dar et al. 2022), or  created a toy example decoder-only LM and used it as their only test-case (Elhage et al. 2021). In particular, this line of work demonstrated GPT-2 is a good proxy for auto-regressive interpretability research.
>
> Thank you for the grammar and style notes, we will fix those.
>
> Please let us know if you have any remaining concerns, especially if you would consider revising your assessment after reviewing our reply.

---

### Meta-Review · Area_Chair_jM1n · 2023-09-11

**Recommendation:** 2

**Metareview:**

This paper presents a tool to visualize a forward pass of GPT-based models as an interactive flow graph, and performs an analysis of the information flow inside the attention mechanism.
It is therefore part demo paper and part interpretability paper. Since it was submitted to the interpretability track, the recommendation is for that track.

There were some concerns around reproducibility, which were (mostly) taken away during discussion, as authors promised to release the code. The paper could still also be improved to provide more details on the approach.

Pros:
- The reviewers unanimously thought the tool to visualize information flow is the biggest contribution of the paper, and so the demo track might have been a better place for it.
- They also do see value in the analysis, especially regarding layer norm.

Cons:
- The analysis is limited to one dataset (CounterFact) and one model (GPT).
- The accessibility of the tool needs improvement.
- The assumption that less activated neurons are less important might not be true.

All in all it's not easy to make a recommendation given the hybrid nature of the paper (part demo paper, part analysis paper). Given that the reviewers quite unanimously positive regarding the above pros and on soundness and excitement, the paper *could* be accepted in current form. But if it is not, I would recommend the authors to either resubmit it as a demo paper or to a workshop, or to make the paper stronger by extending the analysis.

---

### Decision · Program_Chairs · 2023-10-07

**Decision:**

Accept-Findings

**Comment:**

This paper presents a tool to visualize a forward pass of GPT-based models as an interactive flow graph, and performs an analysis of the information flow inside the attention mechanism.
It is therefore part demo paper and part interpretability paper. Since it was submitted to the interpretability track, the recommendation is for that track.

There were some concerns around reproducibility, which were (mostly) taken away during discussion, as authors promised to release the code. The paper could still also be improved to provide more details on the approach.

Pros:
- The reviewers unanimously thought the tool to visualize information flow is the biggest contribution of the paper, and so the demo track might have been a better place for it.
- They also do see value in the analysis, especially regarding layer norm.

Cons:
- The analysis is limited to one dataset (CounterFact) and one model (GPT).
- The accessibility of the tool needs improvement.
- The assumption that less activated neurons are less important might not be true.

All in all it's not easy to make a recommendation given the hybrid nature of the paper (part demo paper, part analysis paper). Given that the reviewers quite unanimously positive regarding the above pros and on soundness and excitement, the paper *could* be accepted in current form. But if it is not, I would recommend the authors to either resubmit it as a demo paper or to a workshop, or to make the paper stronger by extending the analysis.